# RETurn to work After stroKE (RETAKE) Trial: protocol for a mixed-methods process evaluation using normalisation process theory

Kathryn A Radford [1] Christopher McKevitt,[2] Sara Clarke [1] Katie Powers,[1] Julie Phillips,[1] Kristelle Craven,[1] Caroline Watkins,[3] Amanda Farrin,[4] Jain Holmes,[1] Rachel Cripps,[2] Vicki McLellan,[4] Tracey Sach,[5] Richard Brindle,[4] Ivana Holloway,[4] Suzanne Hartley,[4] Audrey Bowen [6] Rory J O'Connor,[7] Judith Stevens,[1] Marion Walker,[1] John Murray,[1] Angela Shone,[8] David Clarke [9]

Professor Walker has now retired and requested not to be included in the publications

For numbered affiliations see end of article.

**Correspondence to**
Professor Kathryn A Radford;
Kathryn.radford@nottingham.ac.uk

## ABSTRACT

**Objectives** This mixed-method process evaluation underpinned by normalisation process theory aims to measure fidelity to the intervention, understand the social and structural context in which the intervention is delivered and identify barriers and facilitators to intervention implementation.

**Setting** RETurn to work After stroKE (RETAKE) is a multicentre individual patient randomised controlled trial to determine whether Early Stroke Specialist Vocational Rehabilitation (ESSVR) plus usual care is a clinically and cost-effective therapy to facilitate return to work after stroke, compared with usual care alone. This protocol paper describes the embedded process evaluation.

**Participants and outcome measures** Intervention training for therapists will be observed and use of remote mentor support reviewed through documentary analysis. Fidelity will be assessed through participant questionnaires and analysis of therapy records, examining frequency, duration and content of ESSVR sessions. To understand the influence of social and structural contexts, the process evaluation will explore therapists' attitudes towards evidence-based practice, competency to deliver the intervention and evaluate potential sources of contamination. Longitudinal case studies incorporating non-participant observations will be conducted with a proportion of intervention and usual care participants. Semistructured interviews with stroke survivors, carers, occupational therapists, mentors, service managers and employers will explore their experiences as RETAKE participants. Analysis of qualitative data will draw on thematic and framework approaches. Quantitative data analysis will include regression models and descriptive statistics. Qualitative and quantitative data will be independently analysed by process evaluation and Clinical Trials Research Unit teams, respectively. Linked data, for example, fidelity and describing usual care will be synthesised by comparing and integrating quantitative descriptive data with the qualitative findings.

**Ethics and dissemination** Approval obtained through the East Midlands—Nottingham 2 Research Ethics Committee (Ref: 18/EM/0019) and the National Health

### Strengths and limitations of this study

► A mixed-methods theory-driven process evaluation will generate detailed findings to assist in interpreting the results of a pragmatic, multicentre individual patient randomised controlled trial of a complex vocational rehabilitation intervention, which crosses the work/health divide.

► This is one of the most comprehensive multisite, multicomponent, multistakeholder perspective process evaluations embedded in a stroke rehabilitation trial, involving detailed assessment of implementation fidelity, therapist competency to deliver the trial intervention, contamination logging and exploration of social and structural influences on intervention provision in poststroke rehabilitation services.

► Longitudinal case studies with intervention and usual care will capture participant experiences of providing and experiencing the intervention including those of employers.

► The COVID-19 pandemic limited researcher access to direct observation of face-to-face intervention delivery and employer interactions with stroke survivors in each site. Integration of interview data from different participant sources, including stroke survivors and carers, occupational therapists and employers with available observational data are planned to address this limitation.

ServiceResearch Authority. Dissemination via journal publications, stroke conferences, social media and meetings with national Stroke clinical leads.
**Trial registration number** ISRCTN12464275.

## BACKGROUND

Approximately 100 000 people in the UK suffer from a stroke every year,[1] and around one in four are of working age.[2] Returning to work after a stroke is a major goal for stroke survivors, contributing to social identity,

emotional and financial well-being and conferring a sense of purpose and has benefits for the individual, the individual's family and the economy.[3] Despite this, only half of working age stroke survivors make a successful return to meaningful work, and they are two to three times more likely to be unemployed 8 years after their stroke than the general population.[1] Although impairments in the stroke survivor's physical, cognitive and communication abilities can affect this,[4 5] social and environmental factors such as personal and employer beliefs and attitudes, job type and organisation size and the benefits system also play an important part.[6 7]

Vocational rehabilitation (VR) is defined as whatever helps someone with a health problem to return to, or remain in, work and includes both work *and* work-related education.[8] It involves helping people find work, helping those who are in work but having difficulty as well as supporting career progression in spite of illness or disability. The primary aim is to optimise work participation.[9] Existing research suggests that VR may help stroke survivors return to their previous job or find new work,[10 11] however, trials to date involve small samples in non-UK settings.

RETurn to work After stroKE (RETAKE) is a multicentre individual patient randomised controlled trial, which aims to determine the clinical and cost-effectiveness of an Early Stroke Specialist Vocational Rehabilitation (ESSVR) intervention in addition to usual National Health Service (NHS) rehabilitation on stroke survivors' return to work (RTW) at 12 months postrandomisation, compared with NHS rehabilitation alone.[12] Acceptability and utility were assessed in a feasibility trial.[13] ESSVR combines conventional occupational therapy (OT) with case coordination. The intervention commences within 2 weeks of randomisation and lasts up to 12 months postrandomisation. It is intended for delivery in the community as often as required by individuals, as determined by a stroke specialist OT with additional VR training. ESSVR includes the following: (a) assessing stroke impact on the person and their job, (b) educating individuals, employers and families about stroke impact on work and strategies to lessen impact (eg, memory aids, fatigue management), (c) work preparation, including opportunities to practice work skills and (d) liaison with employers to plan and monitor a phased RTW (see online supplemental appendix 1). The target number of participants for the trial is 760 participants (420 ESSVR and 340 usual care) from 20 UK hospitals and linked early supported discharge/community services. The RETAKE trial and embedded process evaluation commenced in June 2018 and will complete in March 2022. This period includes a funder approved extension of 7 months necessitated by an unplanned pause in recruitment during the COVID-19 pandemic.

Failure to implement evidence-based stroke rehabilitation interventions in clinical practice may result in unnecessary suffering and disability.[14 15] Trialists must consider future implementation in the real world when designing clinical trials, paying particular attention to the context for intervention delivery and factors likely to influence its uptake and use.[16] This is especially true for trials of complex rehabilitation interventions, which comprise multiple interacting components and target a number of different organisational levels, making them particularly challenging to implement. An embedded process evaluation provides for an in-depth exploration of factors influencing the implementation of complex interventions.

The Medical Research Council (MRC) argue for a systematic approach to designing and conducting process evaluations, drawing on clear descriptions of intervention theory and the identification of key process questions.[17] Mixed methods approaches to process evaluation are increasingly common and consistent with the MRC framework's emphasis on exploring and understanding the important relationship between context, mechanisms and implementation. Theory-driven process evaluations are recommended alongside complex intervention trials to measure what is delivered. These measurements include fidelity (whether the intervention was delivered as intended), dose (the quantity of intervention implemented) and 'reach' of interventions to understand how the intended audience interacts with the intervention.[17] Fidelity data are necessary to interpret intervention outcomes, but despite an extensive literature supporting its importance, fidelity is commonly under-reported in studies of complex rehabilitation interventions. While most trials of VR have not raised particular concerns about fidelity, ESSVR in the RETAKE trial is an example of a particularly complex intervention that crosses organisational boundaries, involves interactions between multiple stakeholders, is highly individually tailored and requires behavioural change by the patient, their family and employer. Therefore, in the process evaluation for the RETAKE trial, we have included specific methods to measure fidelity. Alongside a focus on fidelity, in-depth qualitative exploration of participants' experiences of an intervention and of the social and structural context, in which an intervention is provided, are essential elements of process evaluation of complex interventions. This ensures any adaptations made to tailor intervention to the individual and/or differing contexts, which might undermine fidelity can be evaluated. Understanding and reporting how the intervention (including training and support, communication and management structures) is delivered is important for replication in clinical practice.[17] Such evaluation aims to reduce the chance of discounting effective interventions (type II error) or erroneously attributing outcomes to treatment effectiveness, when interventions are not delivered as intended (type III errors).[18–21] The approach is designed to improve trial design and knowledge translation interventions enhancing clinical implementation and reducing research waste.[22 23]

This paper reports the protocol for the process evaluation embedded in the RETAKE trial.

## AIMS AND OBJECTIVES
### Aims
To measure fidelity to the ESSVR intervention and understand, the social and structural context in which the intervention is delivered and identify factors, which may influence the quality of implementation.

### Objectives
Fidelity measurement and competency assessment will
1. Ascertain intervention dose.
2. Describe content of usual care and ESSVR.
3. Describe levels of adherence to the ESSVR intervention.
4. Understand the delivery of usual Care and ESSVR.
5. Determine OTs competency to deliver ESSVR.
   Social and structural context will include
1. Describe participating sites.
2. Understand professionals' experiences of being trained to deliver the intervention.
3. Understand experiences of delivering the intervention.
4. Understand the social and structural factors, which support or act as barriers to the implementation of the intervention.
5. Understand participants' experience of being supported to RTW after stroke.
6. Identify potential contaminants.

## METHODS
### Design
Embedded theory-driven mixed-methods process evaluation incorporating qualitative and quantitative methods. The process evaluation will draw on the intervention logic model developed by the Trialists (figure 1) and will be underpinned by normalisation process theory (NPT), an implementation theory built on four constructs (coherence, cognitive participation, collective action and reflexive monitoring) each informed by four components.[24] NPT will be used in the development of data collection tools (interview topic guides and observation checklists (see table 1)) and as a sensitising lens in qualitative data analysis and interpretation. NPT constructs will underpin the process evaluation and provide insights into the implementation and integration of the intervention into participating stroke services. This will include how the intervention is received, understood, implemented and how it could be normalised into the current healthcare system.

Column 3 of the logic model identifies the core components of the ESSVR intervention. A more detailed description of the development and feasibility testing of the ESSVR intervention have been published previously.[13]

In addition, the Conceptual Framework for Implementation Fidelity (CFIF) (figure 2) will guide collection and analysis of quantitative data.[25] The CFIF outlines the components and variables that make up and affect intervention fidelity and explains how they relate to each other. Adherence includes content and dose (frequency, coverage and duration) of the delivery.[25]

### Eligibility criteria
Stroke survivors that meet the following criteria for inclusion in the RETAKE trial will be eligible to participate in the process evaluation:
► Age≥18 years.
► Admitted to hospital with new stroke (all severities).
► In work at stroke onset (including self-employed, paid or voluntary).
► Willing and have capacity to provide informed consent to participate in the study.
► Have sufficient proficiency in English to contribute to the data collection required for research.

Potential participants who do not intend to RTW will be excluded. Potential participants with a transient ischaemic attack will be excluded.

Inclusion criteria for carers of potential participants:
► Nominated carer of consenting participant.
► Willing and have capacity to provide informed consent to participate in the study.
► Have sufficient proficiency in English to contribute to the data collection required for research.

### Informed consent
Potential participants will be provided with an information sheet and be provided the opportunity to ask questions of a researcher prior to consent. Written informed consent will be obtained from all participants. When a participant is randomised to the case study element, a researcher will contact the participant to gain consent for interview and observations. Consent will be reaffirmed at the start of interviews. This process will be the same for carer, employer, OT and NHS staff interviews. For employer interviews, additional consent to contact the employer will be requested from the case study participant before the employer is contacted. OTs who will deliver the ESSVR intervention and mentors supporting these OTs will be recruited prior to intervention training. NHS staff involved in the management, commissioning or delivery of stroke rehabilitation in each site participating in the RETAKE trial will be recruited.

### Sampling
For professional and patient interviews, as far as possible, we will use a purposive sampling strategy to ensure diversity in terms of geographical location (eg, urban vs rural centres), level of staff seniority and participant sociodemographic variables (including gender and socioeconomic status). See table 2 for the timepoints at which data collection is planned.

### Patient and public involvement statement
Stroke survivors are involved in all stages of the research cycle.

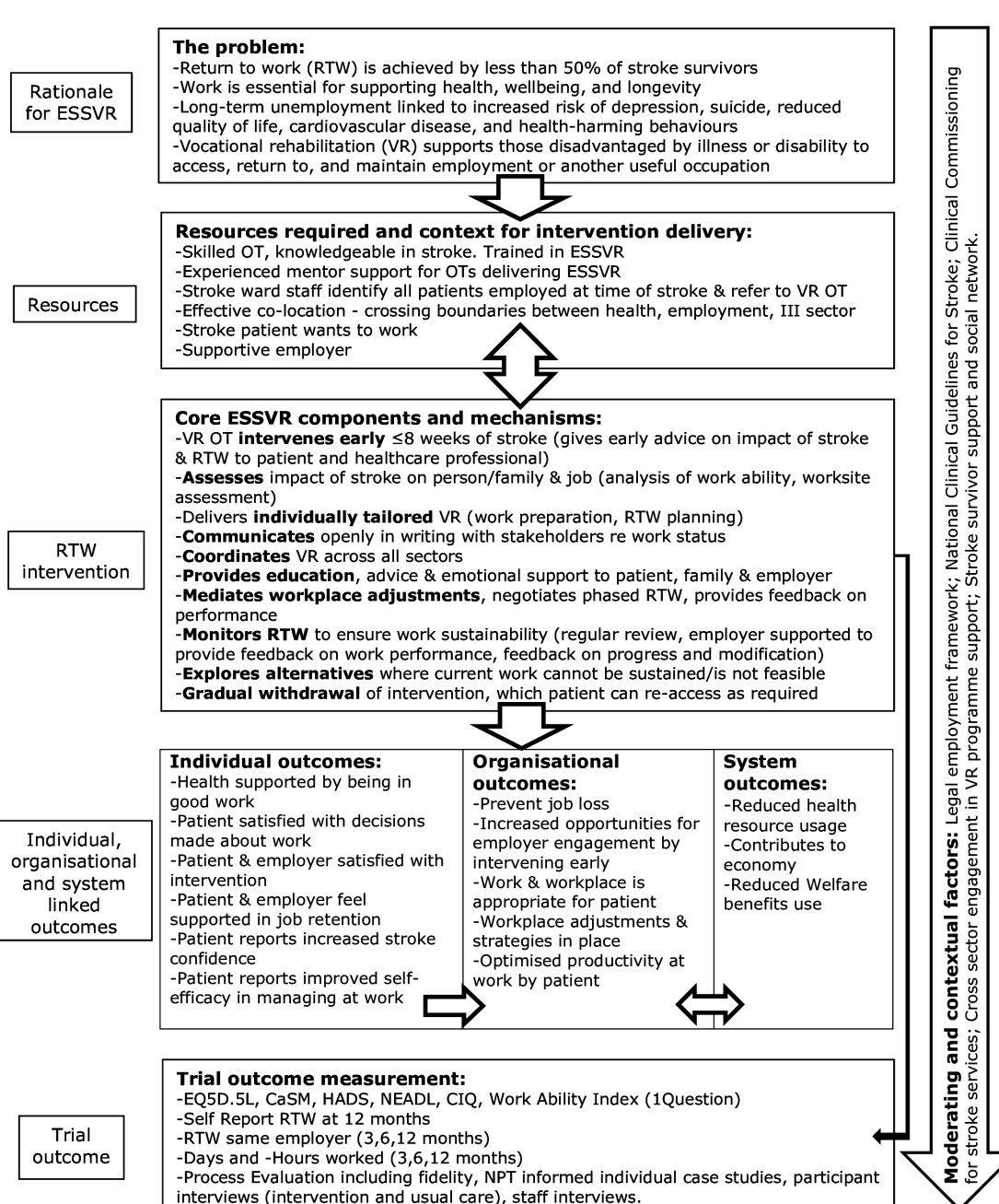

**Early Stroke Specialist Vocational Rehabilitation (ESSVR) Logic Model**
**Aim:** To support patients who have had a stroke to return to and remain in work.

**Rationale for ESSVR**

**The problem:**
-Return to work (RTW) is achieved by less than 50% of stroke survivors
-Work is essential for supporting health, wellbeing, and longevity
-Long-term unemployment linked to increased risk of depression, suicide, reduced quality of life, cardiovascular disease, and health-harming behaviours
-Vocational rehabilitation (VR) supports those disadvantaged by illness or disability to access, return to, and maintain employment or another useful occupation

**Resources**

**Resources required and context for intervention delivery:**
-Skilled OT, knowledgeable in stroke. Trained in ESSVR
-Experienced mentor support for OTs delivering ESSVR
-Stroke ward staff identify all patients employed at time of stroke & refer to VR OT
-Effective co-location - crossing boundaries between health, employment, III sector
-Stroke patient wants to work
-Supportive employer

**RTW intervention**

**Core ESSVR components and mechanisms:**
-VR OT **intervenes early** ≤8 weeks of stroke (gives early advice on impact of stroke & RTW to patient and healthcare professional)
-**Assesses** impact of stroke on person/family & job (analysis of work ability, worksite assessment)
-Delivers **individually tailored** VR (work preparation, RTW planning)
-**Communicates** openly in writing with stakeholders re work status
-**Coordinates** VR across all sectors
-**Provides education**, advice & emotional support to patient, family & employer
-**Mediates workplace adjustments**, negotiates phased RTW, provides feedback on performance
-**Monitors RTW** to ensure work sustainability (regular review, employer supported to provide feedback on work performance, feedback on progress and modification)
-**Explores alternatives** where current work cannot be sustained/is not feasible
-**Gradual withdrawal** of intervention, which patient can re-access as required

**Individual, organisational and system linked outcomes**

**Individual outcomes:**
-Health supported by being in good work
-Patient satisfied with decisions made about work
-Patient & employer satisfied with intervention
-Patient & employer feel supported in job retention
-Patient reports increased stroke confidence
-Patient reports improved self-efficacy in managing at work

**Organisational outcomes:**
-Prevent job loss
-Increased opportunities for employer engagement by intervening early
-Work & workplace is appropriate for patient
-Workplace adjustments & strategies in place
-Optimised productivity at work by patient

**System outcomes:**
-Reduced health resource usage
-Contributes to economy
-Reduced Welfare benefits use

**Trial outcome**

**Trial outcome measurement:**
-EQ5D.5L, CaSM, HADS, NEADL, CIQ, Work Ability Index (1Question)
-Self Report RTW at 12 months
-RTW same employer (3,6,12 months)
-Days and -Hours worked (3,6,12 months)
-Process Evaluation including fidelity, NPT informed individual case studies, participant interviews (intervention and usual care), staff interviews.

**Moderating and contextual factors:** Legal employment framework; National Clinical Guidelines for Stroke; Clinical Commissioning for stroke services; Cross sector engagement in VR programme support; Stroke survivor support and social network.

**Figure 1** The ESSVR logic model. CaSM, Confidence after Stroke Measure; NPT, normalisation process theory. CIQ, Community Integration Questionnaire; EQ5D-5L, EuroQual Five level; HADS, Hospital Anxiety and Depression Scale; NEADL, Nottingham Extended Activities of Daily Living index.

## Design and development

Two stroke survivors are coapplicants on the grant and assisted in identifying the research questions, designing the study and developing the trial protocol.

## Delivery

Two patient and public involvement (PPI) are members of the Trial Steering Committee, and two are members of the Trial Management Group. Additionally, our RETAKE PPI group, which has six members, meets quarterly.

Examples of the work achieved by the PPI group to date are:

► Helping define the primary outcome and defining 'voluntary work', which is included in the definition of the primary outcome.
► Evaluating all patient-facing materials, including aphasia friendly recruitment material.
► Codevelopment of interview topic guides for trial participants and occupational therapists.

**Table 1** Examples of question topics related to NPT constructs

| Normalisation process theory constructs and components | NHS staff/therapist interview topics (some may also arise in informal feedback during training observations) | Stroke participant interview topics (some may also arise in intervention/usual care observations) | Employer interview topics |
|---|---|---|---|
| Coherence:<br>▶ Differentiation<br>▶ Communal specification<br>▶ Individual specification<br>▶ Internalisation | How do staff describe the intervention?<br>How is the intervention similar to/different from usual care?<br>Who would (most) benefit from the intervention? | Experiences of RTW support received: similarities/differences between control and intervention participants | Experience of liaising with the therapist and/or participant on RTW issues |
| Cognitive participation<br>▶ Initiation<br>▶ Enrolment<br>▶ Legitimation<br>▶ Activation | Do staff see value/potential in the intervention?<br>Have they found the training and experience a worthwhile investment of time?<br>Do they feel they have the competence/resources to deliver the intervention effectively? | What were their expectations? Did patients (and carers) value the intervention?<br>How did they respond to the therapists' suggestions?<br>Did they feel they had the ability/resources/confidence to progress through the sessions and ultimately RTW?<br>Context in which participant received RETAKE/acted on suggestions: social, financial, health state, access to opportunities | Expectations of the processes: liaising with therapist/patient and patient's RTW<br>(Prior) experience in supporting RTW for people with disabilities |
| Collective action<br>▶ Interactional workability<br>▶ Relational integration<br>▶ Skill set workability<br>▶ Contextual integration | How compatible is the intervention with the existing stroke care pathway?<br>What other RTW services/resources exist locally? How does this intervention compare/complement those services? Describe working relationships with those services.<br>Support from managers and colleagues during the intervention period | How did participants accommodate the intervention sessions/follow-up actions?<br>How did they manage/are they managing their RTW (if applicable)?<br>Financial implications | Views on who is responsible /roles in supporting RTW<br>Financial implications for example, modifications |
| Reflexive monitoring<br>▶ Systematisation<br>▶ Communal appraisal<br>▶ Individual appraisal<br>▶ Reconfiguration | Perceived effects on patients (and carers)<br>Views on time/resources invested in delivery vs impact<br>What is needed to make it possible to roll out the intervention effectively? (Changes to intervention; changes in services/resources needed for delivery) | Perceived effects of RETAKE/other RTW support<br>Views on time/resources invested in participation vs impact<br>What was good about RETAKE and what could be improved? (Content of intervention sessions/work plans, timing, relationship with therapist) | Perceptions of benefit to employer/tutor/advisor<br>Perceptions of benefit to employee<br>What was helpful about discussions with therapist/participant?<br>What further information/support would they have liked—at what time? |

NHS, National Health Service; NPT, normalisation process theory; RETAKE, RETurn to work After stroKE; RTW, return to work.

▶ Overcoming problems with recruitment. For example, resources and narratives to assist recruiters in approaching people with severe stroke.
▶ Assisting in the design of new materials to promote follow-up, for example, including a 'patient journey leaflet' and Thankyou cards.
▶ Helping reduce the length of follow-up questionnaires.
▶ Advising on communicating with participants during the pandemic.
▶ Changes to the Excess Treatment Cost payment models during trial caused problems for the study. One PPI member wrote directly to Directors of the National Institute for Health Research (NIHR), NHS England, Health and Social Care and the leads for the NIHR Clinical Research Network to explain the impact that these changes on the trial. She received a prompt response, which was extremely helpful to the research team. This has assisted us in explaining the

new system to clinical colleagues and researchers in the Trusts.
▶ Codevelopment of a trial website and trial newsletters.
A draft report on the process evaluation findings will be presented to the PPI group for their consideration and comments prior to submission of the final report to the funder and as part of planning publications and dissemination. The PPI group will be involved in writing up and presenting study findings.

### Data collection
The process evaluation will employ qualitative and quantitative methods to address the research questions. Table 2 illustrates the relationship between the process evaluation aims, research questions, data sources and data collection methods. The following section describes each data source in more detail.

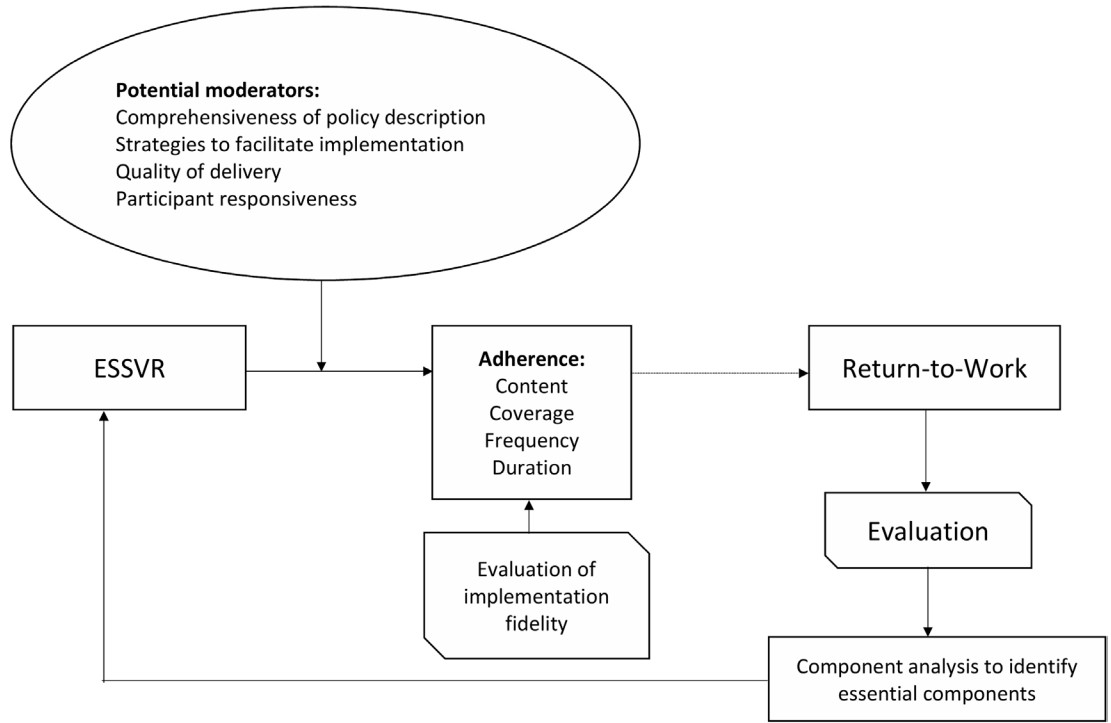

**Figure 2** Assessment of fidelity and factors moderating ESSVR delivery in accordance with the Conceptual Framework for Implementation Fidelity.[25] ESSVR, Early Stroke Specialist Vocational Rehabilitation.

### Intervention content case report forms

To check on fidelity in terms of (early) intervention within 2 weeks of recruitment, initial session case report forms (CRFs) (one per participant) record the intervention start date and whether this occurred within 8 weeks of stroke. Participant summary CRFs record the number of sessions attended out of those proposed and whether there was an agreed ending for the OT-led RTW support. To ascertain intervention dose and describe intervention content, data will be extracted from intervention CRFs for all participants (see table 3). Therapists record each intervention session against predefined components, on the Intervention content CRF.[13] These data will be used to identify which components of the intervention were delivered, to what extent therapists adhered to the intervention process described in the RETAKE manual and to what extent participants adhered to the intervention. For case study participants only, content data will be cross-referenced with the OT's clinical case notes and additional data extracted to explain how the RETAKE intervention interacts with usual care and other services such as employment services. Participants' consent includes permission for members of the trials team to access their therapy records.

### Describing usual care

To describe the content of the intervention and of usual care, resource use questions pertaining to participants' use of health and social care services over the previous 3 months will be completed by all participants at 3, 6 and 12-month postrandomisation as part of follow-up. This data will be used to describe the content of usual care,

and in case study, participants (n=38) will be triangulated with therapists' clinical notes and participant interview transcripts.

### Fidelity

To assess implementation fidelity, a range of data collection methods informed by the CFIF will be used (see table 3).[25]

### Therapist competency assessment

Following attendance at a 2 day, manualised face-to-face training session with VR expert trainers and again at refresher training 6 months later, retake OTs competence will be assessed using OTs written responses to questions based on vignettes depicting novel RTW after stroke scenarios. Model answers developed by the training team will be used to measure competence using criteria based on knowledge of the intervention process (40%), clinical reasoning (50%) and written communication (10%). Scores will be mapped to a rubric identifying OTs as highly competent (≥70%), competent (50%–69%) or needing additional support (≤49%) (see online supplemental appendix 2). In addition, as mentors meet with mentees on a monthly basis, informal monitoring of OT competency can occur. If required, action can be taken to addresses issues of concern identified by mentor or mentee. After 12 months of delivering the intervention, RETAKE OTs competence will be reassessed by evaluating the intervention delivered in a random selection of completed intervention case records (one participant per RETAKE OT) against the trainer's expert opinion. The trainer will review the selected case records against the

**Table 2**  RETAKE process evaluation research questions and data sources

| Aims | Research questions | Data source(s) | Method(s) | Timepoint |
|---|---|---|---|---|
| **Measure fidelity to the intervention** | What is the intervention dose, intensity and duration? | ▶ Intervention content case report forms (CRFs) | Quantitative | Months 3–45 |
| | What is the (reported) content of the ESSVR intervention? What is the content of usual care? | ▶ Intervention content CRFs. ▶ NHS therapy records. ▶ Stroke survivor-reported resource use data. ▶ Stroke survivor carer and OT interviews | Quantitative and qualitative | Months 3–45 Months 12–45 Months 12–36 |
| | Was the intervention delivered with fidelity? What factors affect implementation fidelity? Are RETAKE OTs competent to deliver the ESSVR intervention? | ▶ Fidelity checklist, ▶ Intervention content CRFs ▶ Mentoring records, ▶ RETAKE OT interviews ▶ Individual OT performance in assessed vignettes at baseline and 6 months ▶ RETAKE OT case record reviews at 12 months post training | Quantitative and qualitative Quantitative | Months 3–45 Months 12–18 Months 1–8 and as new OT join the trial and 6 and 12 months post training. |
| **Understand the social and structural context that may influence intervention implementation and future embedding in practice settings.** | What is the context for intervention delivery? | ▶ Site survey at baseline, mid-point and end of intervention delivery | Quantitative and qualitative | Months 1, 18 and 36* *later timepoint for end of intervention delivery where sites recruit beyond the COVID-19 extension. |
| | What services are in place for supporting patients in return to work? | ▶ Site survey at baseline, mid-point and end of intervention delivery | Quantitative and qualitative | As above. |
| | What are the staffing levels at sites? | ▶ Site survey at baseline, mid-point and end of intervention delivery | Quantitative and qualitative | As above |
| | Potential for contamination: Are there proposed or actual VR service developments or changes in practice in place/ planned at site? | ▶ Site survey at baseline, mid-point and end of intervention delivery ▶ NHS staff interviews | Quantitative and qualitative | As above. |
| | What are the RETAKE OTs' perceptions of training and mentoring to deliver the intervention? | ▶ Observations at training sessions ▶ RETAKE OT interviews | Qualitative | Months 1–8 and as new OT join the trial. |
| | How do OTs experience delivering the intervention? | ▶ Observations of ESSVR sessions ▶ RETAKE OT interviews ▶ Mentoring records | Qualitative | Months 12–18 Months 12–18 Months 12–45 |
| | What are the social and structural factors supporting or acting as barriers to intervention implementation? | ▶ Observations of usual care and ESSVR sessions ▶ RETAKE OT interviews ▶ Usual Care therapist interviews ▶ NHS Staff interviews ▶ Mentor interviews | Qualitative | Months 1–8 Months 12–18 Months 12–18 Months 12–24 Months 6–8 |
| | How do participants' experience being supported to return to work after stroke? | ▶ Stroke survivor interviews ▶ Carer interviews ▶ Employer interviews | Qualitative | Months 12–24 Months 12–24 Months 12–24 |

ESSVR, Early Stroke Specialist Vocational Rehabilitation.; RETAKE, RETurn to work After stroKE.

intervention mechanisms identified in the logic model and confirm whether the intervention delivered is consistent with the intervention that would have been delivered by the trainer as an expert RTW-related OT.

Fidelity Checklist

A fidelity checklist based on the RETAKE intervention logic model (see figure 1) and RETAKE intervention process and components will be applied to complete case records (Content of Intervention CRFs, RETAKE OT case notes and Initial Session CRFs) from a random selection of stroke participants randomised to receive the RETAKE

intervention (one per treating RETAKE OT). This will be used in measuring adherence to the RETAKE process and identifying factors affecting adherence.

Mentor interviews and records

Mentoring records

Following training, each treating OT will be assigned a mentor with extensive knowledge and experience of VR. Mentoring will take place monthly via teleconference in small groups (four to six therapists) and serve as an intervention implementation support mechanism. RETAKE OTs will be able to discuss any difficulties they are

**Table 3** CFIF led data extraction for fidelity assessment

| Fidelity measure | CFIF construct* | Measurement tool | Data for extraction | Time point |
|---|---|---|---|---|
| Frequency duration | **Adherence and moderating factors** | **Initial session case report forms (CRFs) Participant summary CRFs** | Intervention start date and end date Number of proposed and attended sessions Whether there was an agreed ending for OT return to work support. | One CRF per participant at Initial session. One CRF per participant completed throughout intervention delivery |
| Intensity (time spent per session) Dose (number of sessions) | **Adherence** | **Intervention content CRF OT clinical records (RETAKE +usual Care)** | Time spent (in minutes) on VR activities per session Description of intervention delivered in each session | One completed following every intervention session In case study participants. |
| Therapist adherence Factors affecting adherence | **Adherence and moderating factors** | **Fidelity checklist** | Components delivered, factors affecting delivery RETAKE process followed Y/N | Applied to one randomly selected completed case per RETAKE OT |
| Real time therapist adherence Factors affecting adherence | **Adherence and moderating factors** | **Mentoring CRFs** | Mentor's concerns about adherence Factors affecting intervention delivery Potential solutions | Completed monthly by mentors |
| Barriers and enablers to intervention delivery | **Moderating factors** | **Interviews with RETAKE therapists** | Factors affecting intervention delivery Potential solutions (developed by OT) | In a random selection of cases during intervention delivery at 3, 6 and 12 months |
| Acceptability of the intervention Barriers and enablers to intervention delivery | **Moderating factors** | **Interviews with stroke participants, carers, employers and NHS staff** | Acceptability of intervention Factors affecting delivery Potential solutions to barriers | Throughout intervention delivery in case studies |

*CFIF adherence includes intervention content, dose, coverage, frequency and duration of intervention; CFIF moderating factors include participant responsiveness, intervention complexity, strategies to facilitate implementation, quality of delivery, recruitment and context.
CFIF, Conceptual Framework for Implementation Fidelity; NHS, National Health Service; RETAKE, RETurn to work After stroKE.

experiencing, ask questions and share best practice with other OTs and their mentor. This process will also facilitate communication between the trial team and enable barriers to implementation and contamination risks to be reported. Key discussion points will be recorded by mentors using a mentoring record form for each session. These records, along with all email correspondence between mentor and mentees, will be collected for qualitative content analysis.

### Mentor interviews
Semistructured interviews will be conducted by two research assistants (SC and KC) with all mentors (n=6) to explore their experiences of supporting RETAKE OTs to deliver the intervention, and ascertain their views of organisational, social and other factors contributing to or affecting delivery of the intervention.

### Social and structural context
#### Site survey
To describe participating sites and identify potential contaminants, sites will be asked to complete a questionnaire by telephone at three time points; prior to

recruitment, halfway through, and at the end of the intervention period. This will contribute to understanding contextual influences through capturing data on existing stroke care pathways and resources (including staff and services) available for supporting participants in a RTW. It will also identify potential contamination risks associated with proposed or planned VR service developments or changes in practice that may influence trial outcomes.

### Therapist training
#### Non-participant observations
To understand OT's experiences of being trained to deliver the intervention, a research assistant (RC) will observe up to four training sessions delivered by the training team. A checklist will be developed using NPT constructs to guide observations. Non-participant observations aim to identify; whether therapists understand the intervention and their role in implementation, whether they think the RETAKE intervention can be integrated into existing practice and any contextual factors affecting the trial.

 Radford KA, *et al. BMJ Open* 2022;**12**:e053111. doi:10.1136/bmjopen-2021-053111

To describe adherence to the intervention, a researcher will observe up to three sessions for each case study participant in the intervention and usual care arms of the trial. Non-participant observations will be conducted using prompts for structured observation and unstructured field notes.[26] Participant selection for inclusion the case study element is described below.

### Interviews with occupational therapists

Semistructured interviews will be conducted by a research assistant (RC) with a minimum of one OT per site following their initial RETAKE training to explore their experience of training, the mentoring process and their confidence in intervention delivery. OT's views of the intervention, barriers and facilitators to implementation and any organisational or social factors impacting on delivery will also be explored. Interviews will take place following training and be repeated at two additional time points: mid-way through the RETAKE intervention delivery and at the end of the study.

### Case studies

Longitudinal case studies will be used to map the care received by RETAKE and usual care participants to develop a more detailed understanding of participants' (stroke survivors, carers, employers) and RETAKE OTs experiences of support for RTW. A 5% subset of participants from both arms of the trial (total n=38) will be randomly selected and invited to participate in the case study element of the process evaluation:

### Case study interviews

Semistructured interviews will be conducted by two research assistants (SC and KC) with case study participants at three time points: 3, 6-month and 12-month postrandomisation, about their experiences and views of and adherence to the RETAKE intervention and the support they received to RTW. The case study participants' carers (if nominated), their employers (where participant consent is obtained) and the OTs providing support for RTW will be interviewed.

### NHS staff interviews

To further understand the social and structural factors which influence the implementation of the intervention, interviews will be conducted with up to two (n=34 in total) NHS staff involved in the management, commissioning or delivery of stroke rehabilitation within each trial site. Participating staff will be chosen using a mixture of purposive and snowball sampling. This will be based on a full range of trial sites, staff knowledgeable about the implementation of the intervention at their site, and staff knowledgeable about the decision-making process relating to wider roll-out.

### Additional participant interviews

An additional random 5% of study participants will be invited to participate in semistructured interviews at the end of the intervention period. These interviews will explore participants' experience of the intervention as well as their perceptions and experiences of returning to work.

All qualitative interviews will be conducted using a topic guide informed by NPT. Examples of question topics and how they relate to the four NPT constructs are shown in table 1. Topic guides will be presented to the RETAKE PPI group for comment prior to use. All interviews will be audio recorded and transcribed in full.

### Data analysis

#### Quantitative analysis

The dose, duration and frequency of the ESSVR intervention will be calculated using data from completed CRFs in combination with NHS therapy records. The total time spent delivering the ESSVR intervention (face to face and non-face to face contact (liaison with the patient, employer and other stakeholders by letter/phone), administration and travel) will be identified. Details relating to the content of intervention sessions will be extracted to identify whether core components of ESSVR were delivered as intended (ie, as specified in the intervention manual and logic model). Associations between therapist attributes, contextual factors and intervention fidelity (measured by deviations from the RETAKE core process) will be explored using regression models. Analysis will be conducted using SPSS (V.21.0 for Windows).

#### Describing usual Care

Data regarding rehabilitation delivered in usual care will be extracted from resource use data in the follow-up questionnaires and from NHS therapy records in case study participants randomised to usual care. These data will be used to inform the cost of usual care for the economic evaluation and describe and understand usual care provided during stroke rehabilitation in inpatient and community services.

Quantitative analysis of these data will be conducted using SPSS (V.21.0 for Windows). Analysis of usual care data obtained from NHS therapy records is described below.

#### Qualitative analysis

Inductive (thematic analysis) and deductive (informed by NPT) approaches will be used to guide data analysis and interpretation. Observational and interview data will be transcribed verbatim and uploaded into QSR NVivo software for management. Descriptions of usual care in NHS therapy records, observational field note data, including researcher reflections and interview data, will be analysed thematically.[27] Framework analysis will be used with the case study data. For each participant, the interview data will be coded in NVivo and then imported into a framework matrix for comparison both within the individual case (comparing views of stroke survivor, carer, OT and employer) and across cases and sites. Analysis will proceed iteratively with data collection to determine whether data saturation has been achieved; researchers will draw on the

RETAKE logic model (figure 1). Throughout the qualitative analysis, NPT will be used as a sensitising framework.

Analysis of each qualitative data set will be conducted independently and then jointly by at least two study team members (SC, KC, KP) to corroborate themes and discuss any discrepancies. It will follow a standard inductive approach of data familiarisation, line-by-line coding and development of broad themes. Themes will then be mapped to NPT constructs as part of development and refinement of broader conceptual explanatory categories. Researchers will keep a set of interim summary notes documenting any reflexivity points and connections between the data with NPT and the logic model, to aid analytical discussions with the wider process evaluation team. Iterative testing of interpretation will occur through discussion with and feedback from the PPI group and discussions within the research team.

## Synthesis of qualitative and quantitative data

During the RETAKE trial, the qualitative and quantitative data generated as part of the process evaluation will be independently analysed by the process evaluation team and the Clinical Trials Research Unit, respectively. Data related to intervention fidelity and description of usual care will be synthesised at the conclusion of the trial. We will review and compare findings from related data sets, identify areas of agreement and disagreement and develop explanations for the findings. Synthesis of findings from both the quantitative and qualitative data generated will contribute directly to the overall evaluation and explanation of the outcomes of the RETAKE trial.

## Ethics and dissemination

Ethics approval has been obtained through the East Midlands—Nottingham 2 Research Ethics Committee (reference 18/EM/0019) and the NHS Research Authority. The procedures for gaining informed consent have been detailed above. Dissemination will be via journal publications, stroke and rehabilitation-focused conferences, newsletter articles, social media, presentations to clinicians and stroke survivors and meetings with national clinical leads for the Stroke Plan and the NHS Plan.

**Author affiliations**
[1]Centre for Rehabilitation and Ageing Research, Faculty of Medicine and Health Sciences, University of Nottingham, Nottingham, UK
[2]Department of Public Health Sciences, School of Population Health and Environmental Sciences, King's College London, London, UK
[3]Faculty of Health and Care, University of Central Lancashire, Preston, UK
[4]Clinical Trials Research Unit (CTRU), University of Leeds, Leeds, UK
[5]Health Economics Group, Norwich Medical School, University of East Anglia, Norwich, Norfolk, UK
[6]Division of Neuroscience and Experimental Psychology, Faculty of Biology, Medicine and Health, University of Manchester, Manchester, UK
[7]Academic Department of Rehabilitation Medicine, Leeds Institute of Rheumatic and Musculoskeletal Medicine, University of Leeds, Leeds, UK
[8]Research and Innovation, University of Nottingham, Nottingham, UK
[9]Academic Unit for Ageing and Stroke Research, Leeds Institute of Health Sciences, University of Leeds, Leeds, UK

**Contributors** KAR, CM, AF, AB, RJOC, MW and CW conceived the study. KAR, DC and CM designed the process evaluation. KAR, CM, DC, SC, KC, JH, JP and KP operationalised the process evaluation protocol. KR, JP, and JH designed the intervention. AS has the role of trial sponsor. IH, RB, and AFa devised the data management and statistical analysis plan. JS and JM acted as PPI collaborators to support plans for trial design/delivery, management, and dissemination of trial findings. VM and SH have responsibility for management of the trial. KAR, SC and DC drafted the manuscript; all other authors read and approved the final version.

**Funding** This study is funded by the NIHR HTA programme (ref: 15/130/11). The views expressed herein are those of the authors, not necessarily the NIHR, the Department of Health and Social Care or the NHS.

**Competing interests** None declared.

**Patient consent for publication** Not applicable.

**Provenance and peer review** Not commissioned; externally peer reviewed.

**ORCID iDs**
Kathryn A Radford http://orcid.org/0000-0001-6246-3180
Sara Clarke http://orcid.org/0000-0001-5919-4282
Audrey Bowen http://orcid.org/0000-0003-4075-1215
David Clarke http://orcid.org/0000-0001-6279-1192

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
