## [Reviewer comments · BMJ Open]

ARTICLE DETAILS

TITLE (PROVISIONAL)	The RETurn to work After stroKE (RETAKE) Trial: protocol for a mixed-methods process evaluation using normalisation process theory
AUTHORS	Radford, Kathryn A; McKeivitt, Christopher; Clarke, Sara; Powers, Katie; Phillips, Julie; Craven, Kristelle; Watkins, Caroline; Farrin, Amanda; Holmes, Jain; Cripps, Rachel; McLellan, Vicki; Sach, Tracey; Brindle, Richard; Holloway, Ivana; Hartley, Suzanne; Bowen, Audrey; O'Connor, Rory J.; Stevens, Judith; Walker, Marion; Murray, John; Shone, Angela; Clarke, David

VERSION 1 – REVIEW

REVIEWER	Katherine Morton University of Southampton, Psychology
REVIEW RETURNED	08-Jul-2021

GENERAL COMMENTS	This is a very detailed and thorough process evaluation, collecting a wide range of in-depth data. My concerns were that only a minimal description of the intervention was provided, whereas more detail could help the reader understand the rationale for the process measures and analysis, and the lack of description about how the findings would be integrated and brought together to achieve the aims. Once these are addressed I think it will be a strong protocol paper. Abstract: Could the researchers say what the purpose of the semi-structured interviews is, and whether/how the qual and quant data will be integrated? Methods: A written description of the intervention components in the methods would really help the reader better understand the process evaluation, or perhaps the RETAKE intervention manual could be included as supplementary material? Given the scale of the process evaluation, it might also help to show visually the different intervention components, and the data being captured at each point? How many participants will be in the full trial? How does the process evaluation fit within the whole trial? When will each source of data collection be completed and analysed? Will the quant data be analysed in parallel with the qual data or sequentially? A figure might help to show this. No dates are included of when the study will begin and end.
---

	Line 206. Are the eligibility criteria the same as those for the RCT? If so, might it be clearer to specify this as the sentence ‘Stroke survivors that meet the following criteria will be considered eligible to participate in the process evaluation’ implied they might be a subset of trial participants? Line 209: does ‘all severities’ of stroke include TIA? Line 214: How is intention to return to work captured? Line 263: Your PPI involvement sounds extremely helpful. I would be interested to know a little more about exactly how PPI will be involved in the process evaluation e.g. I wondered whether the PPI group will be involved in interpreting the process evaluation findings and considering the implications, as well as writing up and presenting findings? Line 277: Why do the initial session CRFs record ‘whether this occurred within 8 weeks of stroke’? Why 8-weeks? Line 299: I wasn’t clear if the vignettes required the OTs to give written or verbal answers. Line 303: Is it possible to share the rubric for assessing OTs’ competence for transparency? Line 307: The criteria for assessing OTs’ competence at 12 months is not very clear. Is it possible to publish key documents, like the intervention content CRF, as Appendices? Line 397: In the case studies, how will the interview data from different participants be analysed, e.g. will you compare data within each case from stroke survivor, carer, OT, employer? Table 4: I found it difficult to relate the participant topics to the NPT constructs, e.g. I wasn’t sure how the support participants report receiving was related to coherence. I wondered if combining Tables 4 and Table 1 might help the reader to see how the NPT definitions can be mapped onto the interview topics. Qual analysis: I wasn’t clear how both inductive and deductive data analyses will be used. You mention NPT will be a sensitising framework throughout, and researchers will draw on the logic model, but is it possible to be more specific? E.g. Will you begin inductively then map your codes and themes to the NPT constructs? What about any data that doesn’t fit with NPT? Will the qual and quant data be analysed separately without integration, or will they be integrated in some way? Perhaps the rationale for your decision about this could be explained. A mixed-methods research checklist might help, such as GRAMMS? Discussion: Is it possible to consider any limitations?
--	--

REVIEWER	Bridget Kiely Royal College of Surgeons in Ireland (RCSI), Department of General Practice
REVIEW RETURNED	21-Jul-2021

GENERAL COMMENTS

Thank you for asking me to review this very thorough and considered process evaluation protocol for a complex intervention. I think the detailed approach to assessing implementation fidelity and the comprehensiveness of the evaluation, underpinned by theory, will be very useful to readers and especially for anyone planning a similar study. The case study approach is very comprehensive, giving a real 360 view over time and including a comparison group of sorts with the non intervention case studies, that will be very interesting. Overall the methods are described in detail, but there could be some clarifications and improvements to how the methods are presented.

I have some queries and suggestions that I hope will improve the clarity, although I appreciate with multiple objectives and data sources it can be challenging to present succinctly.

In the abstract it could be clearer which area (fidelity, social or structural context) each data source is addressing.

In the strengths and limitations I do not note any limitations identified.

Background

There is a minor issue with formatting of references.

Line 113 It would be interesting to know if other VR trials had noted challenges with implementation? In particular around fidelity as you appear to be assessing multiple data sources using both qualitative and quantitative methods for this. Is this because it has been identified as a particular issue in previous studies? The level of monitoring also suggests that this is also part of an effort to maintain high fidelity, so assume any issues with fidelity found during the periodic reviews are going to be addressed as part of the ongoing implementation of the intervention as part of the trial.

Line 152- There is a slight discrepancy between the aim here and the objective stated in the abstract. I would not consider assessing the competency of the OTs as an overall aim as this is listed as an objective, and I think it does sit under a moderating factor of implementation fidelity.

Line 169 Objective 9- Only factors that support the intervention implementation are mentioned here, but I assume you will look at those that hinder it also?

Page 9. There are a number of tables. I wonder if it necessary to include NPT table as it is described and referenced and quite well know. There is also duplication with the table about interview topic guides. Both are possibly superfluous but one would certainly suffice, in which case I would suggest the second with the details about the topic guides.

Figure 1. This is very useful to both describe the intervention and the overall approach, but it is very small. I hope this can be addressed at a formatting stage, otherwise it may need to be condensed further, or a separate figure describing the intervention included.

Methods

P13 line 206 Participants- this reads like eligibility for the main trial.

There are some further details for selection of participants specifically for the process evaluation further below. It could be helpful to outline who the participants for the process evaluation are and how they will be selected and recruited in this section. Also it is a little unclear at what stage OTs and others will consent to take part in the process evaluation? Presumably at the very beginning given the level of observation?

I am impressed by the level of PPI involvement and it is clear how it has impacted the study, in particular engaging with intended beneficiaries.

	Table 2 RETAKE objectives and data sources- While the table is very useful to give an overview of a large number of objectives and data sources, I wonder could it be condensed. For example I feel competency of OTs sits under fidelity and contamination under context as opposed to separate areas. Also you could condense the Social and structural context area. Maybe even just leave it as a broad heading because the detail is in the research questions. I would include participant interviews to answer the question “What are the social and structural factors supporting intervention implementation?” The question for contamination does not quite fit with my understanding of contamination or in the explanation later on page 20 lines 353-356. There is quite a bit of overlap between Table 2 and Table 3 and together they are a bit confusing. Line 285-288. It is unclear where the additional data will come from. Are the OTs keeping their usual notes on top of the trial process evaluation forms? Will case study participants explicitly consented to their records being accessed by members of the trial team? Line 297 Does manualised mean manual based? Line 306 It is not clear how long the intervention will be running for, either individual interventions (although appreciate can only be estimated) or what time period the trial is taking place over. It is a bit unclear how often mentors will be going through the forms. If they are supervising monthly why not have their reviews on adherence more frequently rather than at the seemingly late stage of 12 months when it won't be possible to make any adjustments to improve intervention implementation. Line 309- the Fidelity heading seems a bit misplaced here- should it have been earlier? Line 363-365- Non participant observations- I am not an expert on this method, but I can't see how it would be able to answer the questions suggested. I feel this is a way to measure fidelity of the training to the manual and should be in the fidelity section. The interviews described would address the questions much better. With regards to participant interviews why are you going to do another 5% of participants (ie another 38) interviews. There is already a huge amount of data, with a very comprehensive approach including all stakeholders at a number of different time points in the study. It seems an unmanageable amount and it isn't clear what it would additionally contribute. Line 433-435 ad 427-429 seem to be referring to similar things. What's the difference between the 2 and what different aspects will they assess? Analysis It is unclear how the two approaches are going to fit together. Is the plan to inductively analyse all sources of qualitative data? It would seem a very time consuming approach given the amount of data. Some data sources such as the OT notes and trial forms might be better suited to amore descriptive framework analysis, especially when assessing fidelity? In terms of NPT use in the analysis is the plan to map resulting themes to NPT? Are you planning parallel independent analysis of all data? Again a very detailed approach, but great if you have the time and resources to do it. For the participants checking will that be via the PPI group or directly with participants in the case studies themselves?
--	---

VERSION 1 – AUTHOR RESPONSE

Reviewer 1	
Abstract: Could the researchers say	For the semi-structured interviews an explanation has

what the purpose of the semi-structured interviews is, and whether/how the qual and quant data will be integrated?	been added to lines 42-44 of the manuscript. In relation to plans for integration of the quantitative and qualitative data, a description has been added from lines 46 to 52 of the revised manuscript.
Methods:	
A written description of the intervention components in the methods would really help the reader better understand the process evaluation, or perhaps the RETAKE intervention manual could be included as supplementary material? Given the scale of the process evaluation, it might also help to show visually the different intervention components, and the data being captured at each point?	We have included a sentence in the revised manuscript (lines 228-230) to clarify that the core ESSVR intervention components are listed in column 3 of the logic model. We have reformatted this to make is more easily readable. The intervention manual can not be made available for publication at this time. A more detailed description of the development and feasibility testing of the ESSVR intervention has been published previously. See: Grant et al (2014) https://doi.org/10.4276/030802214X14098207541072 This is reference13 in the revised manuscript. WE also provide a TIDieR description of the intervention in Appendix I Thank you for this suggestion. We have added a column to the revised Table 1 (see reviewer 2 comments below re removing the NPT table and revising the former Table 2), this is designed to show the data being captured in the process evaluation and the different time points at which this was intended to occur . The Covid19 pandemic impacted on recruitment and some of the planned data collection. We will report on the impact of the pandemic on the study in the publication of the findings of the process evaluation and the trial.
How many participants will be in the full trial?	The target number of participants for the trial is 760 participants (420 ESSVR and 340 usual care) from 20 UK hospitals and linked early supported discharge/community services. We have added this information to lines 133 to 135 of the revised manuscript.

How does the process evaluation fit within the whole trial? When will each source of data collection be completed and analysed? Will the quant data be analysed in parallel with the qual data or sequentially? A figure might help to show this.	See response to the question above re a visual representation of when data are collected. With the exception of the mentoring records, the quantitative data are reported on case report forms and returned to the clinical trials research unit (CTRU) for initial analysis. At the end of the trial and following descriptive quantitative analysis in CTRU these data will be shared with the process evaluation team. We will then review and compare these data with the qualitative findings and include these data in the synthesis of the overall findings for the process evaluation. The quantitative data from mentoring records (e.g. frequency and duration of mentoring support) will be collected and reported descriptively by the process evaluation team with copies of these data sent to the CTRU. We have now summarised this information in the abstract and in the manuscript in the data analysis sections.
No dates are included of when the study will begin and end.	We have added this information to lines 135 to 138 of the revised manuscript.
Line 206. Are the eligibility criteria the same as those for the RCT? If so, might it be clearer to specify this as the sentence 'Stroke survivors that meet the following criteria will be considered eligible to participate in the process evaluation' implied they might be a subset of trial participants?	Thank you for highlighting this. The eligibility criteria are the same as for the RCT. We have clarified this point in lines 243 and 244 of the revised manuscript.
Line 209: does 'all severities' of stroke include TIA?	People who have experienced a TIA are not eligible for participation in the trial. Participants need to have been admitted to a stroke unit and have a medically confirmed stroke. We have clarified this at lines 251-252 of the revised manuscript.
Line 214: How is intention to return to work captured?	This forms one of the questions on the consenting process completed at the initial assessment of eligibility for the trial.

Line 263: Your PPI involvement sounds extremely helpful. I would be interested to know a little more about exactly how PPI will be involved in the process evaluation e.g. I wondered whether the PPI group will be involved in interpreting the process evaluation findings and considering the implications, as well as writing up and presenting findings?	The PPI group have two active representatives on the Trial Management Group. They receive and discuss the interim analyses reporting on the ongoing findings of the process evaluation. A draft report on the process evaluation findings will be presented to the PPI group for their consideration and comments prior to submission of the final report to the funder and as part of planning publications and dissemination. We have added some of the above explanation to lines 311—314 of the revised manuscript.
Line 277: Why do the initial session CRFs record 'whether this occurred within 8 weeks of stroke'? Why 8-weeks?	This is to ensure fidelity with the planned early intervention which should commence within 2 weeks of recruitment. We have added to line 327 of the revised intervention to clarify this.
Line 299: I wasn't clear if the vignettes required the OTs to give written or verbal answers.	The OTs provide written responses completed under examination conditions. We have clarified this at line 357 of the revised manuscript.
Line 303: Is it possible to share the rubric for assessing OTs' competence for transparency?	We have included the rubric for assessing competency as an appendix (Appendix II)
Line 307: The criteria for assessing OTs' competence at 12 months is not very clear.	The expert trainer reviews the intervention delivered against the logic model to confirm whether the intervention delivered would have been delivered by that expert OT. We have added clarification of this to lines 368-372 of the revised manuscript.
Is it possible to publish key documents, like the intervention content CRF, as Appendices?	An example of this kind of CRF was published in Grant et al (2014) https://doi.org/10.4276/030802214X14098207541072 This is reference 13 in the revised manuscript.
Line 397: In the case studies, how will the interview data from different participants be analysed, e.g. will you compare data within each case from stroke survivor, carer, OT, employer?	A Framework approach will be used to analyse the case study data. For each participant the interview data is coded in NVivo and then imported into a Framework matrix for comparison both within the individual case

	(comparing views of stroke survivor, carer, OT and employer) and across cases and sites (see lines 515 to 541 of the revised manuscript).
Table 4: I found it difficult to relate the participant topics to the NPT constructs, e.g. I wasn't sure how the support participants report receiving was related to coherence. I wondered if combining Tables 4 and Table 1 might help the reader to see how the NPT definitions can be mapped onto the interview topics.	Thank you for this comment. We have considered this. Please see reviewer 2 comments below re removing the NPT table. We have revised Table 4 (now Table 3) to include the NPT components related to each construct in column 1.
Qual analysis: I wasn't clear how both inductive and deductive data analyses will be used. You mention NPT will be a sensitising framework throughout, and researchers will draw on the logic model, but is it possible to be more specific? E.g. Will you begin inductively then map your codes and themes to the NPT constructs? What about any data that doesn't fit with NPT?	Thank you for this comment, we accept that the concise description of the analysis did not sufficiently communicate the detail of the approaches we are using in the qualitative analysis or the planned integration of the data sets. We have now amended the Data analysis section of the revised manuscript (lines 515-541) to address this.
Will the qual and quant data be analysed separately without integration, or will they be integrated in some way? Perhaps the rationale for your decision about this could be explained. A mixed-methods research checklist might help, such as GRAMMS?	See above comment. We have now commented on the planned integration of the findings from the qualitative and quantitative data generated in the process evaluation in the amended Data Analysis section of the revised manuscript.
Reviewer 2	
In the abstract it could be clearer which area (fidelity, social or structural context) each data source is addressing.	We have added to lines 35-36 and 38-39 to address this comment.
In the strengths and limitations I do not note any limitations identified.	Thank you for pointing out this omission, we have now added one key study limitation.
There is a minor issue with formatting of references.	Thank you. We have taken this to mean the use of commas where reference numbers are consecutive instead of the use of a hyphen. We have corrected the in-

	text citation of references where this had occurred.
Background Line 113 It would be interesting to know if other VR trials had noted challenges with implementation? In particular around fidelity as you appear to be assessing multiple data sources using both qualitative and quantitative methods for this. Is this because it has been identified as a particular issue in previous studies? The level of monitoring also suggests that this is also part of an effort to maintain high fidelity, so assume any issues with fidelity found during the periodic reviews are going to be addressed as part of the ongoing implementation of the intervention as part of the trial.	Thank you for this comment. As we understand it other VR trials have not noted specific issues with fidelity. We have adopted the range of measures outlined in parge part because of the complexity of delivering the ESSVR intervention across service boundaries and over a period of time lasting up to 12 months. We have added text at lines 160-168 to comment on this in the revised manuscript. You are correct in that issues thought to impact on fidelity identified during the monitoring processes in RETAKE will be addressed as part of ongoing implementation.
Line 152- There is a slight discrepancy between the aim here and the objective stated in the abstract. I would not consider assessing the competency of the OTs as an overall aim as this is listed as an objective, and I think it does sit under a moderating factor of implementation fidelity.	We understand your point. We have removed the reference to determining the competence of OTs to deliver the ESSVR intervention from the statement of the aims of the process evaluation, see line 188 of the revised manuscript.
Line 169 Objective 9- Only factors that support the intervention implementation are mentioned here, but I assume you will look at those that hinder it also?	You are correct, it is our intention to explore factors which support implementation and those which act as barrier to intervention implementation. We have amended line 204 in the revised manuscript to reflect this.
Page 9. There are a number of tables. I wonder if it necessary to include NPT table as it is described and referenced and quite well know. There is also duplication with the table about interview topic guides. Both are possibly superfluous but one would certainly suffice, in which case I would suggest the second with the details about the topic guides.	We have considered this comment carefully. We agree that NPT is both well established and well known. We have removed Table 1 as suggested but have retained Table 3 as we feel this illustrates how NPT constructs have informed the selection of questions for the topic guides; we have received feedback previously that this has been regarded as useful by clinicians and some researchers.
Figure 1. This is very useful to both describe the intervention and the overall approach, but it is very small. I hope this can be addressed at a formatting stage, otherwise it may	Thank you for this comment, we liaise with the journal editors to reformat this to make it easier to read.

need to be condensed further, or a separate figure describing the intervention included.	
Methods	
P13 line 206 Participants- this reads like eligibility for the main trial. There are some further details for selection of participants specifically for the process evaluation further below. It could be helpful to outline who the participants for the process evaluation are and how they will be selected and recruited in this section. Also it is a little unclear at what stage OTs and others will consent to take part in the process evaluation? Presumably at the very beginning given the level of observation?	Thank you for highlighting this. The eligibility criteria are the same as for the RCT. We have clarified this point in lines 243 and 244 of the revised manuscript. To clarify this we have added text from lines 268 to 272 and added a section on our approach to sampling from lines 274 to 279. We accept that timepoints for data collection could also have been clearer. We have revised what is now Table 1 (research questions and data sources) to include an additional column in which we have indicated the planned timepoint(s) for data collection. These have been impacted on by the need to seek a funder approved extension to the trial as a result of unplanned pauses in recruitment at sites in 2020/21 due to the Covid19 pandemic.
I am impressed by the level of PPI involvement and it is clear how it has impacted the study, in particular engaging with intended beneficiaries.	Thank you for this comment. PPI is a core value in the Division of Rehabilitation, Ageing and Wellbeing and this group have been important contributors in the study to date.
Table 2 RETAKE objectives and data sources- While the table is very useful to give an overview of a large number of objectives and data sources, I wonder could it be condensed. For example I feel competency of OTs sits under fidelity and contamination under context as opposed to separate areas. Also you could condense the Social and structural context area. Maybe even just leave it as a broad heading because the detail is in the research questions. I would include participant interviews to answer the question "What are the social and structural factors supporting intervention implementation?" The question for contamination does	We understand and accept the points made here. We have revised and reduced the content of the table as far as possible. Participant interviews and potential for contamination now sit under the section on understanding the social and structural context. See the revised Table 1 commencing line 322. Contamination in the RETAKE trial is regarded as any

not quite fit with my understanding of contamination or in the explanation later on page 20 lines 353-356. There is quite a bit of overlap between Table 2 and Table 3 and together they are a bit confusing.	instances where RETAKE OTs have treated usual care participants (sites received funding to allow delivery of the ESSVR intervention by designated RETAKE OTs) or where practice changes to include return to work provision where this had not occurred previously may affect stroke survivors recruited to the usual care arm of the trail. These issues were explored primarily through interviews with OTs, mentors, stroke survivors, carers, and NHS staff. We accept that there is some overlap between what is now Table 1 (formerly Table 2). We feel that Table 2 remains a necessary inclusion given that it primarily focuses on reports how the quantitative data related to fidelity is generated; these data are later compared and then with qualitative interview data as indicated in the table.
Line 285-288. It is unclear where the additional data will come from. Are the OTs keeping their usual notes on top of the trial process evaluation forms? Will case study participants explicitly consented to their records being accessed by members of the trial team?	This is correct, OTs are keeping their usual case notes in addition to recording ESSVR content on CRFs. Case study participants do provide consent for their records to be accessed by members of the research team as you suggest. We have added a sentence to lines 341-342 to confirm this.
Line 297 Does manualised mean manual based?	Yes, the intervention is manual based, but it is not highly protocolised. Therapists need to use their clinical judgement in delivering an individualised intervention based on the core components outlined in the logic model (column 3).
Line 306 It is not clear how long the intervention will be running for, either individual interventions (although appreciate can only be estimated) or what time period the trial is taking place over. It is a bit unclear how often mentors will be going through the forms. If they are supervising monthly why not have their reviews on adherence more frequently rather than at the seemingly late stage of 12 months when it won't be possible to make any adjustments to improve intervention implementation.	We have added a sentence on the duration of the intervention to lines 126 and 127 of the revised manuscript. The timeline for the trail is also clarified in lines 135-138 (this was impacted by the Covid19 pandemic). Re the mentor reviews, as you suggest in practice mentors are informally reviewing the competence of their mentees on a monthly basis and in between monthly meetings where they are asked for individual advice by mentees. Action can be taken if required based on the informal review process or if mentors request this. We have added two sentences at lines 362-365 to clarify this. The reassessment at 12 months is formal trial directed

	process designed to determine changes in competency over time.
Line 309- the Fidelity heading seems a bit misplaced here- should it have been earlier?	We have moved the heading to lines 350-352 in the revised manuscript.
Line 363-365- Non participant observations- I am not an expert on this method, but I can't see how it would be able to answer the questions suggested. I feel this is a way to measure fidelity of the training to the manual and should be in the fidelity section. The interviews described would address the questions much better.	Thank you for this comment. Whilst non-participant observations do provide direct evidence of the extent to which the trainers (mentors) adhered to the intervention manual content, they also provide data on how OTs engaged with and understood the training content and planned implementation. These data arise from formal discussions in the training sessions and from informal comments and discussions occurring during training and during breaks in training. Consent for observations clarifies that these data will be part of the observations and may be recorded in field notes. As you suggest, interviews also explore experiences, perceptions and understanding but at a later time point for the OTs.
With regards to participant interviews why are you going to do another 5% of participants (ie another 38) interviews. There is already a huge amount of data, with a very comprehensive approach including all stakeholders at a number of different time points in the study. It seems an unmanageable amount and it isn't clear what it would additionally contribute.	With hindsight and in principle, we agree with your comment here. The primary purpose of the additional 5% was to capture data from stroke survivors and carers towards the end of the trial in order that any significant changes in intervention delivery or variation by site could be determined. As a team, following the case study analysis and increasing evidence of data saturation we have agreed that the planned number of these interviews can be reduced to 20.
Line 433-435 and 427-429 seem to be referring to similar things. What's the difference between the 2 and what different aspects will they assess?	We agree, this is a duplication and have removed the text from the end of line 499 to 501 in the revised manuscript.
Analysis It is unclear how the two approaches are going to fit together. Is the plan to inductively analyse all sources of qualitative data? It would seem a very time consuming approach given the	The use of inductive approaches will be focused on the analysis of interview and observational data. Please see our responses to the similar questions raised

amount of data. Some data sources such as the OT notes and trial forms might be better suited to a more descriptive framework analysis, especially when assessing fidelity? In terms of NPT use in the analysis is the plan to map resulting themes to NPT? Are you planning parallel independent analysis of all data? Again a very detailed approach, but great if you have the time and resources to do it. For the participants checking will that be via the PPI group or directly with participants in the case studies themselves?	by reviewer 1 and the text added to the abstract at lines 46-52, and to the data analysis section commencing line 515, but to clarify: The qualitative and quantitative data generated during the process evaluation will be independently analysed by the process evaluation team and the Clinical Trials Research Unit respectively. The OT notes and trial forms (CRFs) are being analysed using descriptive content analysis and descriptive statistics in relation to assessing fidelity. This analysis involves the process evaluation team (mentor notes) and the CTRU (CRFs). The CTRU are also responsible for descriptive statistical analysis for all trial data reported in CRFs e.g. dose, duration and frequency of intervention delivery. Directly linked data, for example in relation to intervention fidelity, dose, duration and delivery of ESSVR and description of usual care will be provided to the process evaluation team to support data synthesis at the conclusion of the trial. This will involve comparison and then integration of descriptive findings from the quantitative data with the findings from the qualitative analyses. As indicated previously we used the NPT constructs to develop interview topic guides and to develop the thematic framework for the case study analyses. In terms using NPT in the analyses you are correct that we are conducting the initial qualitative analyses and then mapping the findings to NPT constructs. Re parallel independent analysis: we were fortunate to be able to employ sufficient research assistants and have had the time (in part due to the pandemic) to be able to undertake this kind of analysis. We also had some research assistant turnover; we used the independent parallel analysis to help integrate new research assistants with the more experienced team members. Participant checking is via the PPI group as you suggest.
---	---

VERSION 2 – REVIEW

REVIEWER	Katherine Morton University of Southampton, Psychology
REVIEW RETURNED	25-Jan-2022

GENERAL COMMENTS	Thanks very much for your detailed responses, this is a very interesting study and I look forward to seeing your findings.
--

REVIEWER	Bridget Kiely Royal College of Surgeons in Ireland (RCSI), Department of General Practice
REVIEW RETURNED	28-Jan-2022

GENERAL COMMENTS	Thank you my comments have been addressed. I look forward to the results of the study.
--